# Exploratory analysis of eating- and physical activity-related outcomes from a randomized controlled trial for weight loss maintenance with exercise and liraglutide single or combination treatment

Weight regain after weight loss remains a major challenge in obesity treatment and may involve alteration of eating and sedentary behavior after weight loss. In this randomized, controlled, double-blind trial, adults with obesity were randomized, in a 1:1:1:1 ratio stratified by sex and age group (<40 years and ≥40 years), to one-year weight loss maintenance with exercise, the GLP-1 receptor agonist liraglutide, or the combination, as compared with placebo, after low-calorie diet-induced weight loss. Primary outcome was change in body weight, which has been published. Here, we investigated the effects of weight loss maintenance with exercise, liraglutide, or the combination on weight loss-induced changes in the pre-specified explorative outcomes, eating and sedentary behavior in 130 participants who completed the trial according to the study protocol (exercise ($n = 26$), liraglutide ($n = 36$), combination ($n = 29$), and placebo ($n = 39$)). One year after weight loss, the placebo group had decreased postprandial appetite suppression score by 14%, and increased sedentary time by 31 min/day and regained weight. Liraglutide prevented the decrease in postprandial appetite suppression score compared with placebo (0% vs. −14%; $P = 0.023$) and maintained weight loss. Exercise after weight loss did not increase appetite or sedentary behavior compared with placebo, despite increased exercise energy expenditure and maintained weight loss. The combination of exercise and liraglutide increased cognitive restraint score (13% vs. −9%; $P = 0.042$), reflecting a conscious restriction of food intake, and decreased sedentary time by 41 min/day (−10 vs. 31 min/day; 95%CI, −82.3 to −0.2; $P = 0.049$) compared with placebo, which may have facilitated the additional weight loss. Targeting both eating and sedentary behavior could be the most effective for preventing weight regain.

Trial registration: EudraCT number, 2015-005585-32; clinicaltrials.gov number, NCT04122716.

✉ e-mail: torekov@sund.ku.dk

Weight regain after weight loss represents a major challenge in obesity treatment. Although weight loss can be obtained by calorie restriction, gradual weight regain often occurs, and the original weight is typically reached again within five years[1]. The mechanisms behind the weight regain seem to comprise biological reactions to weight loss, including decreases in resting and non-resting energy expenditure beyond what can be expected based on the actual loss of body mass[2,3] along with changes in appetite-regulating hormones favoring increased food intake[4–6]. However, the actual eating and sedentary behavior may also be important regulators of body weight[7,8]. How different weight loss maintenance strategies affect eating and sedentary behavior has not been characterized.

Obesity is associated with an eating behavior that is characterized by low cognitive restraint, a high degree of emotional and uncontrolled eating, and increased reward responses to energy-dense high-fat foods[9,10]. Together, these features may pose a challenge in weight management.

Sedentary behavior and nonexercise physical activity are important factors of total daily energy expenditure. Decreasing physical activity does not seem to proportionally decrease energy intake[11,12], and a sedentary lifestyle increases the risk of obesity[8,13]. Whether weight loss is associated with a behavioral adaption with decreased non-exercise physical activity is debatable[14,15], but an increase in sedentary behavior and/or decrease in non-exercise physical activity in response to weight loss could impair subsequent weight loss maintenance. Conversely, weight loss could also facilitate physical activity simply by reducing the physical burden.

Short-term studies have demonstrated that glucagon-like peptide-1 receptor agonists (GLP-1RA) induce weight loss by promoting satiety and reducing hunger[16–19]. In a 12-week study, the GLP-1RA, semaglutide, lowered appetite and additionally reduced food cravings and preferences for energy-dense foods[20]. One study lasting for one year showed that the GLP-1RA, liraglutide 3.0 mg, as an adjunct to intensive behavioral therapy, decreased perception of hunger and increased sensation of fullness during the first 24 weeks, but not after one year, despite a greater weight loss achieved with liraglutide[21]. Thus, the role of GLP-1RAs on appetite during long-term maintenance of weight loss is understudied.

Physical activity is recommended in weight loss maintenance strategies to increase energy expenditure and thereby counteract the decrease in energy expenditure otherwise experienced after weight loss. Exercise has been proposed to promote a postprandial 'satiating efficiency' determined as a larger decrease in appetite relative to meal energy content, despite increases in fasting hunger[22]. Furthermore, exercise may reduce liking and wanting for high fat foods[23]. On the other hand, exercise may also be associated with a higher level of appetite and energy intake to match the higher energy requirements of metabolically active tissue[24,25]. However, the effects of exercise on appetite after weight loss have not been clarified.

We have previously shown that a diet-induced weight loss of 13 kg was maintained with exercise or liraglutide as single treatments and that body weight was further reduced with the combination of both treatments, in contrast to weight regain with placebo after 1 year[26]. An investigation of the mechanisms of action of these treatment modalities seems crucial for the improvement of strategies for long-term weight loss maintenance. Thus, the aim of the present study was to investigate changes in appetite, eating and sedentary behavior, and non-exercise physical activity during one-year weight loss maintenance with moderate-to-vigorous intensity exercise, liraglutide 3.0 mg, or a combination of both, compared with placebo after an initial diet-induced weight loss.

Here, we show that weight loss is associated with both increased appetite and sedentary time during one-year weight maintenance. Treatment with a GLP-1RA prevented the increase in appetite and maintained weight loss. Exercise after weight loss did not lead to a greater increase in appetite than the placebo group, despite increased exercise energy expenditure and maintained weight loss. The combination of exercise and GLP-1RA improved both cognitive restraint, reflecting a conscious restriction of food intake and prevented the increase in sedentary behavior, and resulted in additional weight loss. Thus, targeting both eating and sedentary behavior seems the most effective for preventing weight regain.

## Results

### Study population

To characterize appetite, eating, and sedentary behavior after weight loss, we obtained measures from participants who completed a one-year weight maintenance study preceded by an initial weight loss induced by a low-calorie diet of 800 kcal/day for eight weeks. After the initial weight loss, participants were randomly assigned to a one-year weight loss maintenance phase with either a moderate-to-vigorous intensity exercise program plus placebo (exercise group), liraglutide 3.0 mg/day (liraglutide group), a combination of exercise and liraglutide (combination group), or placebo (placebo group). Eligible participants were adults (age 18-65 years) with obesity (BMI 32-43 kg/m$^2$) without diabetes (see the full list of eligibility criteria here[27]). A total of 195 participants completed the eight-week low-calorie diet, of whom 166 participants underwent assessments after one year. Of these, 130 participants adhered to their designated treatment arm (per-protocol population) and were included in the present analyses. A flow diagram of participants in the study is provided in Figure S1. At inclusion, participants' mean BMI was $36.9 \pm 2.9$ kg/m$^2$, mean age was $45 \pm 12$ years, and 62% were women (Table 1). Mean sedentary time per day was $10.9 \pm 1.5$ hours/day. The low-calorie diet induced a weight loss of 13.7 kg. One year after randomization, in the per-protocol population, the placebo group had regained 6.1 kg. Weight loss was maintained in the exercise group (0.7 kg; difference from placebo, −5.3 kg; 95% CI, −9.4 to −1.2) and liraglutide group (−1.9 kg; difference from placebo, −8.0 kg; 95% CI, −11.7 to −4.2), and a further reduction of −6.0 kg was observed in the combination group (difference from placebo, −12.1 kg; 95% CI, −16.1 to −8.1).

### Weight loss is associated with increased appetite and sedentary time that persist after one year

We collected ratings of prospective food consumption, hunger, fullness, and satiety on a visual analog scale in fasted state and during a three-hour mixed meal test to investigate changes in appetite during the study. These ratings of appetite were combined in an overall appetite suppression score (calculated as satiety + fullness + [100−hunger] + [100−prospective food consumption])/4[17]) with

## Table 1 | Characteristics of participants before and after the low-calorie diet

| | Before weight loss | After weight loss, at randomization |
|---|---|---|
| | All (n = 130) | All (n = 130) |
| Male/female (n) | 50/80 | 50/80 |
| Age (years) | 45 ± 12 | 45 ± 12 |
| BMI (kg/m$^2$) | 36.9 ± 2.9 | 32.4 ± 2.9 |
| Body weight (kg) | 110.2 ± 14.7 | 96.5 ± 12.3 |
| Sedentary time (min/day) | 656 ± 92 | 637 ± 91 |
| Cognitive restraint score | 38.8 ± 18.0 | 42.8 ± 16.2 |
| Emotional Eating score | 47.6 ± 27.9 | 42.9 ± 27.2 |
| Uncontrolled Eating score | 52.7 ± 18.0 | 46.2 ± 19.2 |
| OAS score | 10300 ± 3389 | 10488 ± 3412 |

OAS overall appetite suppression.
Values are observed means ± SD for study participants who adhered to the study interventions (per-protocol population).

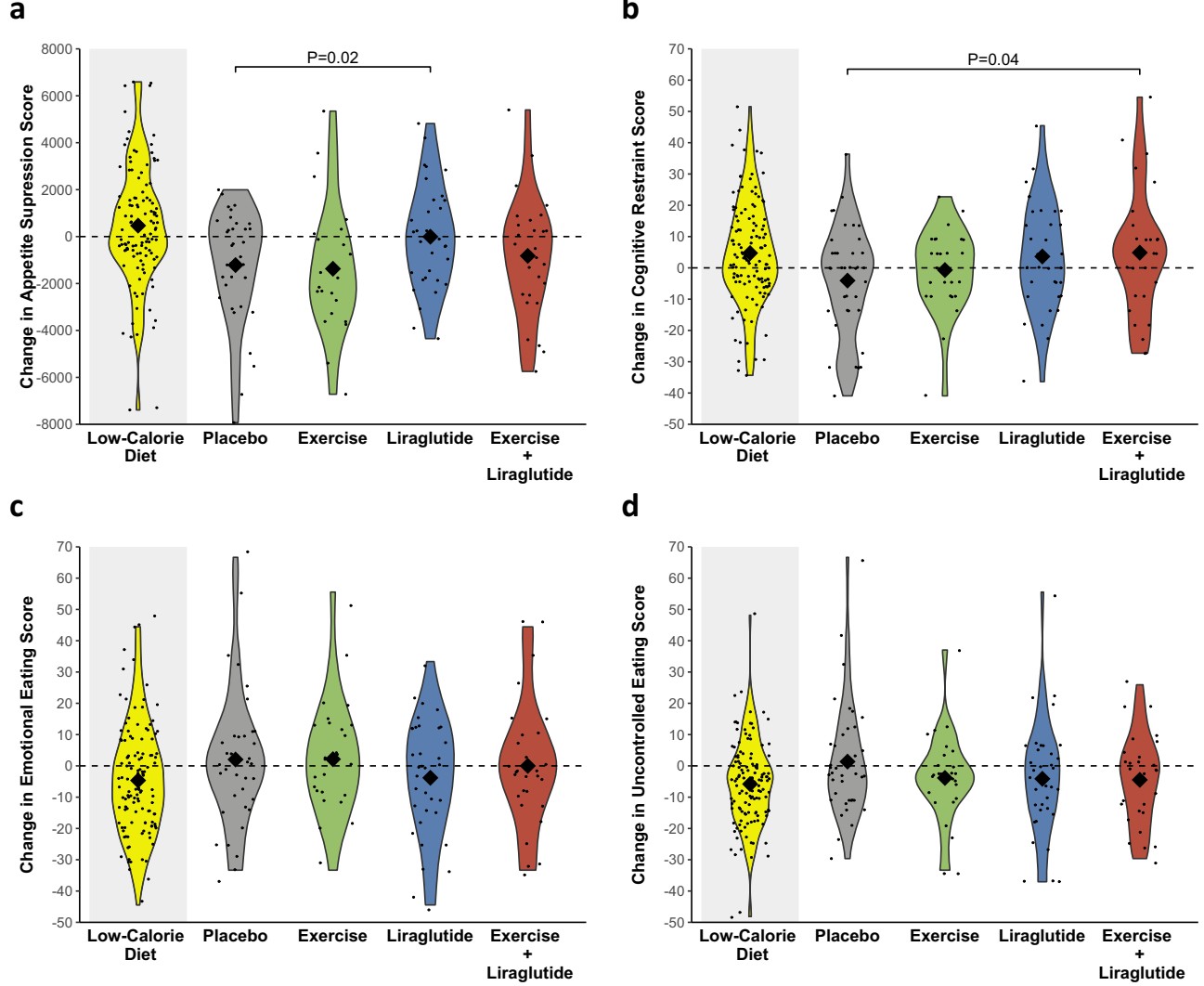

**Fig. 1 | Changes in postprandial appetite suppression and eating behavior during the study.** Violin plots of observed changes in **a** postprandial appetite suppression score, **b** cognitive restraint, **c** emotional eating, and **d** uncontrolled eating. Changes are from week −8 to week 0 (low-calorie diet, yellow color in grey shaded area, $n = 130$) for all groups combined and from week 0 to week 52 (after randomization) in the four groups, separately: placebo (grey color, $n = 39$), exercise (green color, $n = 26$), liraglutide (blue color, $n = 36$), and the combination of liraglutide and exercise (red color, $n = 29$). The diamonds indicate observed means and the dots indicate individual observations. The appetite suppression score was calculated as (satiety + fullness + [100 − hunger] + [100 − prospective food consumption])/4; all were subjective ratings (180 min area under the curve from visual analogue scales. Cognitive restraint, emotional eating, and uncontrolled eating scores were measured on a 0–100 scale by Three-Factor Eating Questionnaire-R18. Results are presented for study participants who adhered to the study interventions (per protocol population). *P*-values are provided for significant (*P* < 0.05) differences from the placebo group. P-values are two-sided and unadjusted for multiple comparisons and derived from a mixed model analysis as described in *statistical analyses*. Source data are provided as a Source Data file.

increasing values indicating more appetite suppression. To study the natural development of appetite after weight loss, we analyzed changes from after weight loss to one year after weight loss in the placebo group, who received no active treatment other than regular weight consultations and dietetic support in accordance with the recommendations from the Danish Authorities. After one year, the placebo group experienced increases in appetite to an extent that were higher than before the weight loss. Increases were observed in postprandial ratings of prospective food consumption (19.2%; *P* = 0.01) and hunger (21.9%; *P* = 0.002) whereas decreases were observed for ratings of postprandial fullness (−13.4%; *P* = 0.01) and satiety (−13.6%; *P* = 0.008). Accordingly, the postprandial overall appetite suppression score decreased in the placebo group (−14.0%; *P* = 0.001) (Fig. 1a and Table 2).

To investigate changes in habitual physical activity and sedentary behavior after weight loss, participants wore accelerometer devices on their wrists for seven consecutive days before and after the diet-

induced weight loss and at weeks 13, 26, and 52 of weight maintenance. Accelerometer data showed that moderate-to-vigorous intensity physical activity was stable during the low-calorie diet (Table 3). During weight loss maintenance, the placebo group did not show changes in any of the physical activity parameters, however, daily sedentary time was increased by 31 min (*P* = 0.03) (Fig. 2). In support of this finding, when participants were asked how much time they spent sitting in a day on average, the placebo group reported a mean increase of 25 min/day from after weight loss to after one year (Table 3).

### The GLP-1RA liraglutide counteracts the increased appetite during weight loss maintenance

Liraglutide treatment prevented the increases which were observed in the placebo group after one year with regards to postprandial ratings of prospective food consumption (−2.0% vs. 19.2%; *P* < 0.05) and hunger (−1.7% vs. 21.9%; *P* = 0.02) (Table 2, S1). Accordingly, the postprandial overall appetite suppression score was maintained with

**Table 2 | Changes in eating behavior and appetite parameters during low-calorie diet and weight loss maintenance by intervention group**

| | LOW-CALORIE DIET | WEIGHT MAINTENANCE PHASE | | | |
|---|---|---|---|---|---|
| | All (*n* = 130) | Placebo (*n* = 39) | Exercise (*n* = 26) | Liraglutide (*n* = 36) | Liraglutide + Exercise (*n* = 29) |
| Male/female (n) | 50/80 | 15/24 | 11/15 | 13/23 | 11/18 |
| Age (years) | 45 ± 12 | 44 ± 12 | 45 ± 12 | 46 ± 10 | 44 ± 13 |
| **EATING BEHAVIOR** | | | | | |
| Cognitive restraint score (no unit) | 4.2 (1.2; 7.2) | −3.8 (−9.3; 1.7) | −0.7 (−7.4; 6.0) | 3.4 (−2.3; 9.1) | 4.9 (−1.5; 11.2) |
| Emotional eating score (no unit) | −4.7 (−8.3; −1.1) | 1.8 (−4.6; 8.1) | 2.1 (−5.6; 9.8) | −3.8 (−10.4; 2.8) | 0.0 (−7.3; 7.3) |
| Uncontrolled eating score (no unit) | −6.7 (−9.3; −4.0) | 1.3 (−4.0; 6.6) | −3.8 (−10.3; 2.6) | −4.1 (−9.6; 1.5) | −4.5 (−10.6; 1.7) |
| **SUBJECTIVE APPETITE RATINGS (FASTING)** | | | | | |
| PFC (mm) | −6.9 (−10.7; −3.2) | 1.0 (−4.9; 7.0) | 3.2 (−4.0: 10.3) | −0.8 (−6.8; 5.2) | 4.0 (−2.6; 10.6) |
| Hunger (mm) | −0.4 (−4.4; 3.5) | −4.4 (−12.3; 3.5) | 0.9 (−8.6; 10.4) | −5.5 (−13.5; 2.6) | −1.6 (−10.5; 7.2) |
| Fullness (mm) | −2.7 (−6.3; 0.9) | 0.1 (−6.5; 6.7) | 2.8 (−5.1; 10.7) | 3.6 (−3.1; 10.3) | 4.8 (−2.6; 12.2) |
| Satiety (mm) | 1.9 (−2.0; 5.8) | −2.9 (−9.3; 3.6) | −3.1 (−10.8; 4.6) | 8.1 (1.5; 14.6) | −1.6 (−8.8; 5.5) |
| OAS score (mm) | 1.6 (−1.3; 4.6) | 0.1 (−4.9; 5.1) | −1.1 (−7.1; 4.9) | 4.5 (−0.5; 9.6) | 0.2 (−5.4; 5.8) |
| **SUBJECTIVE APPETITE RATINGS (POSTPRANDIAL)** | | | | | |
| PFC (180 min x mm) | −485 (−1012; 43) | 1445 (573; 2317) | 1407 (342; 2472) | −155 (−1039; 728) | 858 (−118; 1834) |
| Hunger (180 min x mm) | −326 (−874; 222) | 1612 (627; 2597) | 1497 (297; 2696) | −116 (−1113; 881) | 1330 (228; 2433) |
| Fullness (180 min x mm) | 273 (−255; 800) | −1262 (−2222; −302) | −1024 (−2198; 150) | −293 (−1267; 681) | −502 (−1578; 575) |
| Satiety (180 min x mm) | 370 (−162; 903) | −1310 (−2275; −346) | −1354 (−2531; −177) | −116 (−1093; 862) | −661 (−1742; 419) |
| OAS score (180 min x mm) | 366 (−102; 835) | −1407 (−2236; −578) | −1326 (−2337; −315) | −34 (−873; 806) | −837 (−1765; 91) |
| **FOOD PREFERENCES** | | | | | |
| High fat sweet, Explicit liking (mm) | −8.7 (−12.0; −5.4) | −2.5 (−8.1; 3.1) | 0.5 (−6.5; 7.5) | −6.1 (−12.0; −0.3) | 1.3 (−5.1; 7.9) |
| High fat sweet, Implicit wanting (no unit) | −9.8 (−14.1; 5.4) | 6.7 (1.5; 12.0) | 7.2 (0.6; 13.7) | 3.0 (−2.5; 8.5) | 2.3 (−3.8; 8.4) |
| High fat savory, Explicit liking (mm) | −0.7 (−3.9; 2.6) | −6.9 (−12.9; −1.0) | −4.8 (−12.2; 2.6) | −11.2 (−17.4; −5.0) | −2.2 (−9.2; 4.7) |
| High fat savory, Implicit wanting (no unit) | 7.7 (4.0; 11.3) | −6.5 (−12.3; −0.7) | −8.0 (−15.2; −0.9) | −11.3 (−17.3; −5.3) | −5.6 (−12.3; 1.0) |

*OAS* overall appetite suppression, *PFC* prospective food consumption.
Results are presented for study participants who adhered to the study interventions (per-protocol) as estimated mean changes (95% CI) during the low-calorie diet for all groups combined and for the four groups separately from randomization to week 52. Values were estimated from a linear mixed model with time, group, sex, age, and a time group interaction as fixed effects.

**Table 3 | Changes in sedentary time and physical activity during low-calorie diet and weight loss maintenance by intervention group**

| | LOW-CALORIE DIET | WEIGHT MAINTENANCE PHASE | | | |
|---|---|---|---|---|---|
| | All (*n* = 130) | Placebo (*n* = 39) | Exercise (*n* = 26) | Liraglutide (*n* = 36) | Liraglutide + exercise (*n* = 29) |
| **ACCELEROMETRY** | | | | | |
| Sedentary time (min/day) | −12 (−27; 2) | 31 (3; 59) | 8 (−24; 40) | 15 (−17; 47) | −10 (−40; 20) |
| Light-intensity PA (min/day) | −7 (−16; 2) | 5 (−11; 22) | −1 (−19; 18) | −13 (−31; 6) | 24 (6; 41) |
| MVPA[a] | 0.99 (0.93; 1.06) | 1.04 (0.92; 1.18) | 1.04 (0.90; 1.19) | 0.98 (0.85; 1.12) | 1.20 (1.05; 1.37) |
| **SELF-REPORTED** | | | | | |
| Sitting time (min/day) | −7 (−39; 29) | 25 (−28; 77) | −14 (−83; 56) | −8 (−64; 48) | −98 (−161; −35) |
| Walking[a] | 1.21 (1.00; 1.47) | 0.84 (0.58; 1.21) | 0.84 (0.53; 1.32) | 1.13 (0.76; 1.68) | 0.90 (0.57; 1.40) |
| MVPA[a] | 1.10 (0.84; 1.45) | 1.24 (0.82; 1.89) | 1.57 (1.01; 2.46) | 0.88 (0.58; 1.32) | 1.36 (0.88; 2.10) |

*PA* physical activity, *MVPA* moderate-to-vigorous intensity physical activity.
Results are presented for study participants who adhered to the study interventions (per-protocol) as estimated mean changes (95% CI) during the low-calorie diet for all groups combined and for the four groups separately from randomization to week 52. Values were estimated from a linear mixed model with time, group, sex, age, and a time group interaction as fixed effects.
[a]Values are estimated geometric mean ratios (below one is a decrease and above one is an increase).

liraglutide contrasting a reduction in the placebo group (−0.3% vs. −14.0%; *P* = 0.02) (Fig. 1a). Fasting appetite ratings were generally unchanged throughout the weight maintenance phase, except fasting satiety that increased in the liraglutide group compared with the placebo group (24.7% vs. −9.4%; *P* = 0.02). The appetite-suppressing effects of liraglutide were also apparent in the analyses of the intention-to-treat population (Fig. S3 and Table 4, S2).

Participants were provided with a computerized task, the Leeds Food Preference Questionnaire[28,29], in order to measure changes in preferences for four food categories: high-fat sweet, low-fat sweet, high-fat savory, and low-fat savory. Explicit liking was calculated, indicating the extent to which participants liked each food item. Implicit wanting was calculated as an indication of preference for one food type compared with other food types, with response time included in the algorithm so that a positive score would indicate a more rapid implicit preference for a specific food type. After one year, explicit liking for high-fat sweet foods was decreased by 16.1% in the liraglutide group (*P* = 0.04). A decrease in implicit wanting for high-fat

**a**

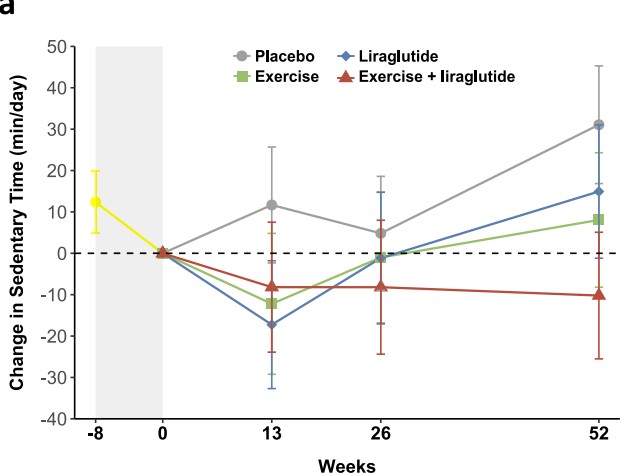

**b**

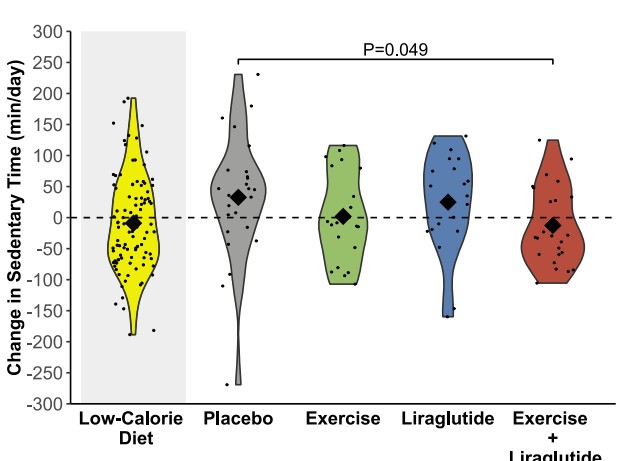

Fig. 2 | **Changes in sedentary time during the study. a** Mean±SEM changes in sedentary time from week −8 to week 0 (low-calorie diet, yellow color in grey shaded area, n = 130) for all groups combined and from week 0 to week 52 (after randomization) in the four groups, separately: placebo (grey color, n = 39), exercise (green color, n = 26), liraglutide (blue color, n = 36), and the combination of liraglutide and exercise (red color, n = 29). Means were estimated from a mixed model analysis as described in *statistical analyses*. Results are presented for study participants who adhered to the study interventions (per protocol population).

Sedentary time was measured with wrist-worn accelerometry. **b** Violin plots of observed changes in sedentary time the low-calorie diet and for all groups, separately. The diamonds indicate observed means and the dots indicate individual observations. Results are presented for the per protocol population (n = 130). *P* values are provided for significant (*P* < 0.05) differences from the placebo group. *P*-values are two-sided and unadjusted for multiple comparisons and derived from a mixed model analysis as described in *statistical analyses*. Source data are provided as a Source Data file.

sweet foods was observed during the initial weight loss, and this was maintained in the liraglutide group but increased in the placebo group (28%; *P* = 0.01). The liraglutide group also showed decreased explicit liking and implicit wanting for high-fat savory foods (*P* < 0.001 for both). Sedentary time and physical activity measures did not change in the liraglutide group.

**Exercise increases appetite similar to placebo despite increased exercise energy expenditure and maintained weight loss**

In general, the amount of energy expended during exercise was maintained throughout the study period, as shown in Fig. 3a. After the initial six-week ramp-up phase, the mean extra energy expended on planned exercise was estimated to be 1661 ± 738 kcal/week in the exercise group and 1473 ± 711 kcal/week in the combination group. Exercise duration and intensity were 156 ± 54 min/week at 78 ± 4% of maximum heart rate in the exercise group and 144 ± 67 min/week at 78 ± 5% of maximum heart rate in the combination group (Fig. 3b, c). In the exercise group, the overall appetite suppression score decreased by 12.1%, similar to the placebo group (*P* = 0.9), despite the extra energy expended. Implicit wanting for high-fat sweet foods increased in the exercise group (35%; *P* = 0.03) to a similar extent as the placebo group. Thus, participants who exercised experienced increased appetite after weight loss. However, this was not more than the placebo group, and despite the increased exercise energy expenditure in the exercise group that maintained weight loss compared with weight regain in the placebo group.

**The combination of liraglutide and exercise improves cognitive restraint and reduces sedentary time during weight maintenance**

In addition to the assessment of appetite, we applied questionnaires to assess participants' changes in eating behavior. From the three-factor eating questionnaire-R18[30], we calculated scores on cognitive restraint, emotional eating, and uncontrolled eating. Cognitive restraint refers to a conscious restriction of food intake to control body weight or lose weight. Uncontrolled eating refers to overeating with a feeling of losing control over food intake. Emotional eating refers to overeating in

response to negative emotions (e.g., feeling lonely or anxious)[30]. Each score reflects the tendency toward the behavior in question, with a higher score indicating a greater tendency towards the specific behavior. One year after randomization, the combination group exhibited a higher cognitive restraint score than the placebo group (12.8% vs. −8.6%; *P* = 0.04). The liraglutide group also tended to have a higher cognitive restraint score than the placebo group (8.3% vs. −8.6%; *P* = 0.07), while the exercise group did not differ from the placebo group (*P* = 0.47) (Table 2 and Fig. 1b). Emotional eating and uncontrolled eating scores were not significantly changed in any of the groups (Table 2 and Fig. 1c, d). In the combination group, the change in appetite ratings was not significantly different from placebo (change in overall appetite suppression score, −7.9% vs. −14.0%; *P* = 0.37).

Sedentary time was decreased by 10 min/day in the combination group, in contrast to the 31 min/day increase with placebo (difference, −41 min/day; *P* = 0.049). In support of this finding, participants' self-reported time spent sitting was lowered in the combination group compared with the placebo group (*P* = 0.003) (Table 3, S1). In the intention-to-treat population (including all randomized participants regardless of adherence), the self-reported daily sitting time was lower in the combination group compared with placebo, whereas the accelerometer measured sedentary time was not significantly decreased compared with placebo (Table 4, S2). This discrepancy was driven by increased sedentary time in some participants randomized to the combination group who did not adhere to the treatment regimen (Fig. S4).

**Increased cognitive restraint and moderate-to-vigorous intensity physical activity are associated with maintained weight loss**

We performed a multiple regression analysis in order to test if changes in the investigated behaviors were associated with changes in body weight during the weight maintenance phase (Table 5). Increased cognitive restraint score and amount of moderate-to-vigorous intensity physical activity were associated with less weight regain during weight maintenance (*P* = 0.003 and *P* = 0.02, respectively). Increased uncontrolled eating score tended to be associated with regain of weight (*P* = 0.07).

**Table 4 | Changes in outcomes during low-calorie diet and weight loss maintenance by intervention group in the intention-to-treat population**

| | LOW-CALORIE DIET | WEIGHT MAINTENANCE PHASE | | | |
|---|---|---|---|---|---|
| | (n = 195) | Placebo (n = 49) | Exercise (n = 48) | Liraglutide (n = 49) | Liraglutide + Exercise (n = 49) |
| EATING BEHAVIOR | | | | | |
| Cognitive restraint score (no unit) | 5.3 (3.0; 7.7) | −4.6 (−9.6; 0.4) | −1.0 (−6.1;4.1) | 3.1 (−1.8; 8.0) | 1.3 (−3.5; 6.0) |
| Emotional eating score (no unit) | −3.7 (−6.5; −0.8) | 1.4 (−4.5; 7.4) | 0.2 (−5.9; 6.3) | −3.8 (−9.6; 2.1) | −1.8 (−7.4; 3.9) |
| Uncontrolled eating score (no unit) | −6.1 (−8.1; −4.1) | 1.5 (−3.4;6.5) | −4.5 (−9.5;0.5) | −2.9 (−7.7; 2.0) | −3.7 (−8.3; 1.0) |
| SUBJECTIVE APPETITE RATINGS (POSTPRANDIAL) | | | | | |
| PFC (180 min x mm) | −453 (−878; −29) | 1283 (435; 2132) | 1382 (531; 2233) | −10 (−818; 799) | 1007 (216; 1798) |
| Hunger (180 min x mm) | −168 (−622; 287) | 1469 (527; 2412) | 1344 (400; 2288) | −121 (−1018; 776) | 1332 (455; 2211) |
| Fullness (180 min x mm) | 478 (31; 925) | −1261 (−2203;−318) | −1138 (−2082; −194) | −359 (−1259; 541) | −1081 (−1962;−199) |
| Satiety (180 min x mm) | 486 (55; 918) | −1247 (−2187; −306) | −1318 (−2260; −377) | −284 (−1179; 612) | −1122 (−1998;−246) |
| OAS score (180 min x mm) | 395 (9; 782) | −1322 (−2128; −516) | −1310 (−2117; −503) | −122 (−890; 646) | −1154 (−1905; 403) |
| FOOD PREFERENCES | | | | | |
| High fat sweet, Explicit liking (mm) | −9.2 (−12.0; −6.5) | −2.1 (−7.6; 3.3) | 2.1 (−3.6; 7.7) | −6.1 (−11.5; −0.7) | 1.1 (−4.1; 6.3) |
| High fat sweet, Implicit wanting (no unit) | −9.4 (−12.7; 6.1) | 7.0 (1.8; 12.3) | 8.1 (2.6; 13.6) | 2.3 (−2.9; 7.4) | 2.7 (−2.3; 7.7) |
| High fat savory, Explicit liking (mm) | −2.1 (−4.9; 0.7) | −5.7 (−11.5; 0.1) | −4.7 (−10.7; 1.3) | −11.4 (−17.1; −5.7) | 0.5 (−5.0; 6.1) |
| High fat savory, Implicit wanting (no unit) | 6.8 (3.9; 9.8) | −6.9 (−12.7; −1.1) | −8.5 (−14.5; −2.5) | −10.2 (−15.9; −4.5) | −3.3 (−8.8; 2.2) |
| ACCELEROMETER-DERIVED PHYSICAL ACTIVITY | | | | | |
| Sedentary time (min/day) | −7 (−20; 6) | 34 (4; 63) | −2 (−30; 27) | 11 (−20; 42) | 21 (−7; 49) |
| Light-intensity PA (min/day) | −7 (−14; 0) | 4 (−13; 20) | −1 (−17; 14) | −8.0 (−25; 9) | 14 (−1; 29) |
| MVPA[a] | 0.99 (0.93; 1.04) | 1.03 (0.91; 1.17) | 1.05 (0.93; 1.18) | 1.00 (0.88; 1.14) | 1.08 (0.96; 1.21) |
| SELF-REPORTED PHYSICAL ACTIVITY | | | | | |
| Sitting time (min/day) | 5 (−25; 34) | 33 (−21; 88) | −17 (−76; 43) | −39 (−92; 15) | −54 (−107; 0) |
| Walking[a] | 1.04 (0.89; 1.23) | 0.85 (0.60; 1.21) | 0.99 (0.69; 1.42) | 1.18 (0.82; 1.69) | 1.03 (0.73; 1.46) |
| MVPA[a] | 1.06 (0.87; 1.29) | 1.11 (0.77; 1.61) | 1.53 (1.09; 2.14) | 0.81 (0.57; 1.15) | 1.31 (0.94; 1.82) |

*PFC* prospective food consumption, *OAS* overall appetite suppression, *MVPA* moderate-to-vigorous intensity physical activity.

Results are presented for all randomized participants (intention-to-treat population) as estimated mean changes (95% CI) during the low-calorie diet for all groups combined and for the four groups separately from randomization to week 52. Values were estimated from a linear mixed model with time, group, sex, age, and a time group interaction as fixed effects.

[a]Values are estimated geometric mean ratios (below one is a decrease and above one is an increase).

## Harms

Harms in the study have been reported in detail elsewhere[26]. In brief, gastrointestinal adverse events were more frequent in the liraglutide and combination group than in the exercise and placebo groups. This included nausea, diarrhea, and vomiting. The number of participants reporting decreased appetite during the trial was 2 (4%) in the placebo group, 4 (8%) in the exercise group, 18 (37%) in the liraglutide group, and 16 (33%) in the combination group. A total of 16 (8%) serious adverse events were reported, of which 5 (3%) led to discontinuation of study medication (3 in the exercise group, 1 in the liraglutide group, and 1 in the combination group).

## Discussion

We investigated changes in appetite, eating, and sedentary behavior during weight loss maintenance with liraglutide, exercise, or the two combined, compared with a placebo group. Placebo treatment after weight loss was associated with increased appetite and sedentary time, which may have contributed to the weight regain after weight loss. Liraglutide treatment after weight loss prevented the increase in appetite, which may have contributed to the maintained weight loss. Exercise after weight loss was not associated with increases in appetite compared with placebo, despite increased exercise energy expenditure and maintained weight loss. Compared with placebo treatment, the combination of exercise and GLP-1 RA improved cognitive restraint score, reflecting a conscious restriction of food intake, and prevented the increase in sedentary behavior, which may have facilitated the additional weight loss.

After weight loss, the perception of postprandial hunger, prospective food consumption, and wanting of food items that were sweet and high in fat increased in the placebo group, while the perception of postprandial satiety and fullness decreased. Increased appetite promotes increased energy intake and weight regain[4,31], and is therefore likely to have contributed to a positive energy balance and thereby the weight regain of almost 50% of the initial weight loss within one year observed in the placebo group. These results support that weight loss induces an increase in appetite[6,7,32] that persists for at least one year and seems to complicate adherence to lifestyle changes and cause weight regain. Some studies have shown a decrease in physical activity during calorie restriction[14,33]. In our study, there were no acute changes in physical activity during the low-calorie diet and a non-significant change in sedentary time of −12 min/day. However, an increase in sedentary time of 31 min/day in the period after weight loss was observed in the placebo group, which persisted for at least one year after weight loss. Thus, increased sedentary time may contribute to weight regain after weight loss.

In the liraglutide group, the increased appetite observed in the placebo group was prevented, and the diet-induced weight loss was maintained with a treatment effect of −8.0 kg versus placebo. Liraglutide treatment decreased postprandial ratings of hunger and prospective food consumption and increased satiety in the fasted state compared with placebo. These findings concur with the described mode of action of GLP-1RAs, i.e., promoting satiety and inhibiting appetite[16,17,20], and extend this notion to weight loss maintenance for at least one year. As reflected in the overall appetite suppression score,

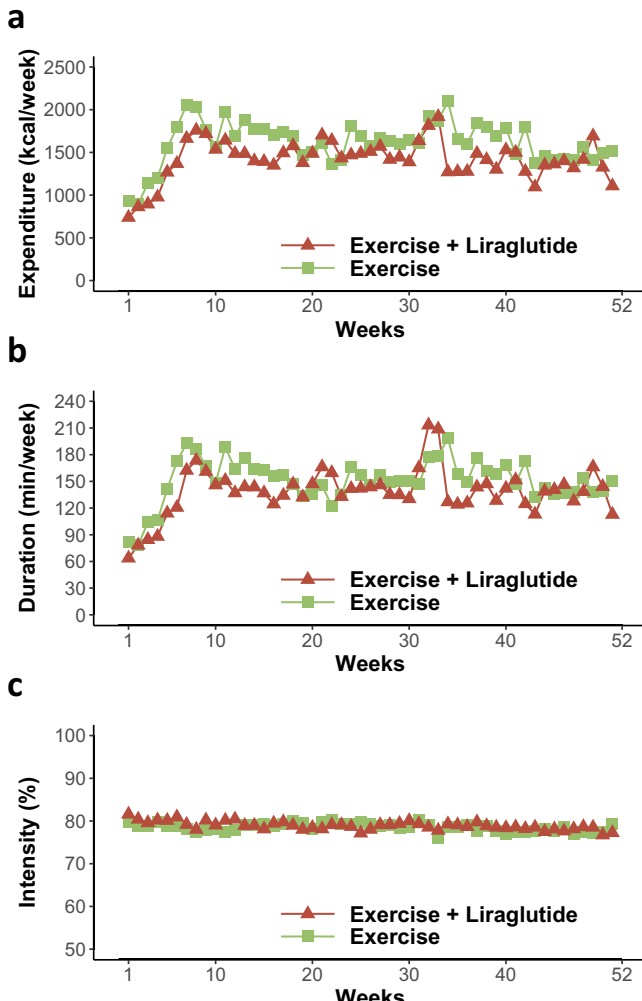

**Fig. 3 | Performed exercise during the study. a** Weekly energy expended during exercise (kcal/week), **b** weekly exercise duration (min/week), and **c** weekly exercise intensity (percentage of maximum heart rate) from the first week after randomization (week 1) throughout the one-year intervention period in the two groups that included an exercise intervention: exercise (green color, $n = 26$) and the combination of exercise and liraglutide (red color, $n = 29$). The exercise program consisted of a gradual increase from week 1 to 6 after which the exercise program was fully implemented. Plotted values are observed means for study participants who adhered to the study interventions (per protocol population). Source data are provided as a Source Data file.

increased in the exercise and combination group, in agreement with data from the sports watches. Weight loss was maintained in the exercise group with a treatment effect of −5.3 kg compared with placebo[26]. Increased moderate-to-vigorous intensity physical activity was associated with less weight regain in the whole study population, in support of physical activity being important for weight maintenance. The estimated increase in energy expenditure of exercise was 1661 kcal/week in the exercise group throughout the intervention period. Despite this increase in energy expenditure, no increase in sedentary time was observed compared with the placebo group. Furthermore, appetite ratings were similar in the exercise group compared with the placebo group. Thus, increased overall physical activity in the exercise group was not associated with a further increase in appetite compared with placebo, which, together with the increased energy expenditure from the exercise, may have contributed to successful weight maintenance.

In the combination group, the estimated increase in energy expenditure of exercise was 1473 kcal/week throughout the intervention period. Despite this exercise-induced increased energy expenditure, the increased sedentary time as seen in the placebo group after weight loss was prevented in the combination group, with the combination group being 41 min/day less sedentary compared to the placebo group. The combination group also improved the cognitive restraint score, reflecting an intent to restrict energy intake to control body weight. Performing linear regression analysis on the study group as a whole, we found that increased cognitive restraint score was associated with less weight regain during weight maintenance (Table 5). Cognitive restraint has previously been proposed to be an important factor in weight loss maintenance[9,34]. The combination group had a substantial weight reduction of −12.1 kg compared with placebo[26], which might in itself increase appetite[6]. This could explain why the decreased appetite as observed with liraglutide alone was not observed in the combination group. Notably, although not significant, the combination group experienced changes in the measured appetite outcomes that ranged somewhere between those in the placebo and the liraglutide groups. Furthermore, the previously published safety profile of this study showed that decreased appetite was reported frequently during the study and to a similar degree in the liraglutide group (37%) and combination group (33%) compared with only 2% in the placebo group[26]. Thus, it is likely that the increase in appetite associated with further weight loss and increased energy expenditure from the exercise program was actually prevented with GLP-1RA stimulation.

Strengths of this study include the randomized, placebo-controlled design, the novelty in focusing on weight loss maintenance after a diet-induced weight loss, and that the intervention period was of one-year duration, investigating eating and sedentary behavior before and after weight loss and during weight maintenance. It is a potential limitation that we cannot prove causal inferences that the measured outcomes are the reason for the weight changes, although this seems probable. We have presented both the intention-to-treat population and the per-protocol population, the latter meaning those who completed the study and adhered to the allocated treatment. The per-protocol analysis reflects the effects when the treatments are taken optimally, and we did this to get a better mechanistic understanding of the behavioral changes that occur after weight loss and how the actual performed treatments may affect these. A limitation of this approach is the introduction of potential subset selection bias that cannot be distinguished from the treatment effect[35]. However, we also performed the analyses on the full data set including all randomized participants (intention-to-treat population). In total, 36 participants completed the study without adequate adherence to allocated treatment. Including these participants in the analyses (Table 4) generally showed the same picture as that presented for the per-protocol population.

the appetite inhibition with liraglutide was more pronounced post-prandially than in the fasted state. In the liraglutide group, the preference for high-fat foods was reduced after one year, indicating that GLP-1RA treatment also affects hedonic aspects of eating behavior on a long-term basis. These findings are in line with observations made after 12 weeks of treatment with the GLP-1RA semaglutide 1.0 mg/week, showing lower liking and wanting of high-fat, energy-dense foods[20], assessed with a similar test as the one used in the present study. Sedentary time or physical activity did not appear to be affected by GLP-1RA treatment. Together, the durable effect of liraglutide in maintaining the diet-induced weight loss seems to involve inhibition of the increased appetite that was observed in the placebo group, which persisted for at least a year.

The use of sports watches with heart rate monitors showed that the exercise program was successfully implemented and that exercise was performed at a high intensity throughout the trial in both the exercise and the combination group (Fig. 3). Questionnaire data confirmed that moderate-to-vigorous intensity physical activity was

**Table 5 | Associations between changes in eating and physical activity parameters and weight change**

| Dependent variable: Percentage change in body weight | | | |
|---|---|---|---|
| | Unstandardized β (95% CI) | Standardized β | *P* value |
| ΔCognitive restraint | −0.17 (−0.27 to −0.06) | −0.30 | **0.003** |
| ΔUncontrolled eating | 0.13 (−0.01 to 0.26) | 0.19 | 0.070 |
| ΔEmotional eating | −0.03 (−0.12 to 0.06) | −0.07 | 0.538 |
| ΔOAS score | 0.00010 (−0.00058 to 0.00079) | 0.027 | 0.765 |
| ΔSedentary time | −0.006 (−0.029 to 0.017) | −0.057 | 0.621 |
| ΔMVPA | −0.078 (−0.140 to −0.015) | −0.276 | **0.016** |

*OAS* overall appetite suppression, *MVPA* moderate-to-vigorous intensity physical activity.

Data are presented as unstandardized and standardized regression coefficients (β) (95% CI) for all randomized participants (intention-to-treat population). Regression coefficients and *P*-values were estimated from a multiple regression analysis. Change in body weight (%) was included as the dependent variable and age, gender, study group, and baseline body weight was included as covariates in the model.

*P* values are two-sided and unadjusted for multiple analyses. *P* values in boldface are statistical significant (<0.05).

In summary, placebo treatment after weight loss was associated with increased appetite and sedentary time, which may have contributed to the weight regain after weight loss. These results support the importance of initiating an active weight maintenance treatment after weight loss, targeting appetite, sedentary behavior, and, preferably, both. Liraglutide treatment after weight loss prevented the increase in appetite, which may have contributed to weight loss maintenance. Exercise after weight loss did not lead to a greater increase in appetite than the placebo group, despite increased exercise energy expenditure and maintained weight loss. The combination of exercise and liraglutide improved cognitive restraint and decreased sedentary behavior compared with placebo, which may have facilitated the additional weight loss. Thus, targeting both eating and sedentary behavior seems the most effective approach for preventing weight regain after weight loss.

## Methods

### Study design

The reported results were part of a randomized placebo-controlled, 2-by-2 factorial trial (EudraCT number, 2015-005585-32; clinicaltrials.gov number, NCT04122716). The study protocol, statistical analysis plan, and primary outcome (body weight) have been published[26,27]. The study was approved by the Regional Ethics Committee for the Capital Region of Denmark (H-16027082) and the Danish Medicines Agency and was carried out in accordance with the principles of the Declaration of Helsinki and ICH Good Clinical Practice guidelines. All study participants provided written informed consent before enrollment in the study. Participants who completed the study received a minor compensation (3000 Danish kroner (DKK) before tax) for time used in the trial during working hours.

### Participants

Men and women (*n* = 215) were included from August 29, 2016, to September 14, 2018, at Hvidovre University Hospital and Department of Biomedical Sciences, University of Copenhagen, Denmark. The last participant's last visit was November 28, 2019. Participants were adults (age 18–65 years) with obesity (BMI 32–43 kg/m$^2$) without diabetes (see the full list of eligibility criteria here[27]).

### Description of interventions

All participants underwent an initial eight-week low-calorie diet of ~800 kcal/day. Participants with weight loss ≥5% (*n* = 195) were then randomly assigned, in a 1:1:1:1 ratio (with stratification according to gender and age group (<40 years and ≥40 years), to a one-year weight loss maintenance phase with either exercise+placebo, liraglutide, a combination of exercise and liraglutide, or placebo. Assignment of participants to treatment was performed by a study nurse based on a randomization list provided by Novo Nordisk.

The starting dose of liraglutide (or volume-matched placebo) was 0.6 mg/day with 0.6 mg weekly increments until 3.0 mg/day or the highest dose at which participants did not have unacceptable adverse events. The participants and the investigators were blinded with respect to liraglutide or placebo treatment until the analyses of the primary outcome were complete. Participants randomized to exercise underwent a seven-week ramp-up phase with increasing exercise duration before being encouraged to attend supervised group exercise sessions (30 min of vigorous-intensity indoor cycling and 15 min of circuit training) two times per week and to perform exercise individually at a moderate-to-vigorous intensity two times per week. Participants in the placebo and liraglutide groups were instructed to maintain usual physical activity throughout the trial.

### Outcomes

All outcomes were measured before and after the eight-week low-calorie diet and after one year of weight maintenance treatment.

### Eating behavior

A Danish version of the three-factor eating questionnaire-R18[30] was used to assess eating behavior based on three factors: cognitive restraint, emotional eating, and uncontrolled eating. Cognitive restraint refers to a conscious restriction of food intake to control body weight or lose weight. Uncontrolled eating refers to overeating with a feeling of losing control over food intake. Emotional eating refers to overeating in response to negative emotions (e.g., feeling lonely or anxious)[30]. Each of the 18 questions belongs to one of the three factors. The questions were rated on a scale from one to four, reflecting the tendency towards the behavior in question. The sum of questions belonging to a specific factor was transformed to a 0–100 scale ((raw score − lowest possible raw score)/possible raw score range) × 100) with a higher score indicating a greater tendency towards the specific behavior[36].

### Appetite

After an overnight fast, participants were instructed to consume a standardized liquid mixed meal over a time interval of 15 min. The liquid meal consisted of two nutritional drinks (Nutricia Nutridrink) containing 600 kcal (74 g carbohydrates, 23 g fat, 24 g protein). Prospective food consumption, hunger, fullness, and satiety were measured on a 100 mm visual analog scale (VAS) with the most extreme statements at each end[37]. VAS measures were obtained in fasted state before ingestion of the liquid meal and at time points 15, 30, 45, 60, 90, 120, 150, and 180 min. Postprandial ratings were calculated as total area under the curve using the trapezoid rule. An overall appetite suppression score was calculated as (satiety + fullness + [100−hunger] + [100−prospective food consumption])/4[17].

### Food preferences

Participants were provided with a culturally adapted Danish version of the validated Leeds Food Preference Questionnaire (LFPQ)[28,29] in the fasted state. Participants were presented with pictures of food items from four categories (high-fat sweet, low-fat sweet, high-fat savory, and low-fat savory) from which components of food preference and reward (conscious/explicit liking and subconscious/implicit wanting) were assessed. Pictures of food items are available in Fig. S2. Food images were presented one by one for the participants to rate the extent to which they liked each food item (i.e., "How pleasant would it be to taste this food now?") using a 100 mm VAS to measure explicit liking. Measurement of implicit wanting was performed by pairing images of each of the four food categories to every other category. Participants were instructed to answer as quickly as possible to indicate preference, i.e., "Which food do you want to eat the most now?". Reaction times were used to calculate the mean response time for each food category (adjusted for frequency of selection). A frequency-weighted algorithm was used to account for selection and non-selection, also included in the rating[29]. A total score that was positive would indicate a more rapid implicit preference for that food type compared with other food types, whereas a negative score indicated the opposite.

### Exercise energy expenditure

Sports watches with chest strap heart rate monitors (Polar A300, Polar Electro Oy, Kempele, Finland) were worn during all exercise sessions. Based on sex, age, height, weight, maximum heart rate, and heart rate during exercise, energy expenditure during exercise was extracted using algorithms in the device and summarized for each week after randomization.

### Sedentary behavior and physical activity

Daily sedentary time and time spent on light- and moderate-to-vigorous intensity physical activity were measured objectively with triaxial wrist-worn accelerometers (GENEActiv, Activinsights Ltd., Camridgeshire, UK) worn by the participants for seven consecutive days on five occasions: before and last week of the low-calorie diet, and at week 13, 26, and 52 of the weight maintenance phase. The sampling frequency was set to 75 Hz. Accelerometer data were processed using the R-package "GGIR" v1.11[38]. This processing included automatic calibration[39], detection of sustained abnormally high values, detection of non-wear time[40], and calculation of the average magnitude of acceleration expressed as Euclidean norm minus one (vector magnitude of acceleration minus gravity) averaged for five-second epochs with negative values rounded to zero. Files were included in the analysis if data were available for >16 h/day on at least three days of the week, as this has been shown to be representative of a full week[41]. Time spent sedentary, in light-intensity, moderate-intensity, and vigorous-intensity physical activity was calculated using cut-off intervals as described elsewhere[40,42]. Since the structured exercise program was mostly performed on stationary bikes, wrist-worn accelerometry was not expected to detect this as moderate-to-vigorous activity. Thus, we also applied a Danish version of the International Physical Activity Questionnaire – Short Form (IPAQ-SF)[43] as a subjective measure of sedentary time and physical activity to get a more thorough description of overall physical activity in the study. IPAQ-SF is a 7-item questionnaire assessing the time spent sitting, walking, and on moderate-intensity and vigorous-intensity physical activity during the past seven days.

### Statistical analyses

The sample size was calculated based on change in body weight, which was the primary outcome and has been published[26]. It was estimated that at least 30 participants in each group would be necessary to detect a clinical relevant 4 kg difference between any of the four treatment groups. In this paper, we report results on pre-specified secondary outcomes. Changes in outcomes were analyzed using a linear mixed model with the following fixed effects: time, group, sex, age group (<40 years vs. ≥40 years), and a time-group interaction. An unstructured covariance pattern and a repeated effect for visit on participant-level were included in the model. We performed the analyses on both the per-protocol and intention-to-treat population. The per-protocol population includes participants with adequate adherence to study interventions, and was predefined (see[26]) for the exercise intervention as meeting at least 75% of WHO's recommendations on physical activity for health in adults: at least 150 min of moderate-intensity aerobic physical activity, or 75 min of vigorous-intensity aerobic physical activity, or an equivalent combination of both[44]. For study medication, per-protocol was defined as having administered 2.4 or 3.0 mg/day subcutaneous liraglutide/placebo for at least 75% of the intervention period. The intention-to-treat population was defined as all randomized participants regardless of adherence. For all analyses, missing data was assumed to be missing at random. Potential reasons for missingness are presented in Fig. S1. For data that did not meet the assumption of normality of residuals, statistical analyses were performed on log-transformed data and back-transformed for presentation (as ratios with 95% CI). Multiple linear regression analysis was performed to assess potential associations between changes in the investigated outcomes and change in body weight during the weight loss maintenance intervention. Age, gender, intervention group, and baseline body weight were included as covariates in the model. The analyses were exploratory and unadjusted for multiplicity, wherefore definite inferences cannot be made. All significance testing was performed using $\alpha = 0.05$. Statistical analyses were performed using SAS Enterprise Guide v7.15 (SAS Institute Inc., Cary, NC, USA).

### Reporting summary

Further information on research design is available in the Nature Research Reporting Summary linked to this article.

## Data availability

The study protocol and statistical analysis plan have been published[26,27]. Source data are provided with this paper. De-identified data under general data protection regulations (GDPR) may be available for research collaboration purpose upon reasonable request to the corresponding author (Signe Sørensen Torekov, torekov@sund.ku.dk), and will require the completion of a data processing agreement. Source data are provided with this paper.

## Code availability

No previously unreported custom computer code was used in this manuscript.

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

## Acknowledgements

We would like to thank all the study participants for their participation in the S-LITE trial. Conduction of the trial would not have been possible without help from pre-graduate students and other staff at the Hvidovre University Hospital. The study was supported by an Excellence grant from the Novo Nordisk Foundation (NNF16OC0019968, received by SST), by a grant from the Novo Nordisk Foundation (NNF15CC0018486, Tripartite Immunometabolism Consortium, received by SST), by Helse-fonden, the Danish Diabetes Academy (received by CJ), and the Faculty of Health and Medical Sciences, University of Copenhagen (received by JRL). Novo Nordisk supplied study medication, and Cambridge Weight Plan supplied Cambridge diet products and accelerometers. The funders of the study had no role in study design, data collection and analysis, or manuscript writing.

## Author contributions

S.B.K.J. and C.J. contributed equally to the study. S.S.T., B.M.S., J.J.H., S.M., J.R.L., C.J., and S.B.K.J. designed the trial. C.J., S.B.K.J., J.R.L., L.M.O., C.R.J., R.M.C., A.A., S.A.B., and I.C.J. contributed to the practical execution of the trial. C.J. and S.B.K.J. processed accelerometer data and performed the statistical analyses. G.F. and C.J. established and adapted the LFPQ. S.B.K.J., C.J., R.M.C., and S.S.T. drafted the manuscript with contributions on methodology from A.A. and S.A.B. All authors contributed, edited, and approved the manuscript.

## Competing interests

SM: *Advisory boards*: AstraZeneca; Boehringer Ingelheim; Eli Lilly; Merck Sharp & Dohme; Novo Nordisk; Sanofi Aventis. *Lecture fees*: AstraZeneca; Boehringer Ingelheim; Merck Sharp & Dohme; Novo Nordisk; Sanofi Aventis. *Research Grant Recipient*: Novo Nordisk, Boehringer-Ingelheim. JJH: Advisory boards: Novo Nordisk. SST: *Research Grant Recipient* Novo Nordisk. RMS: Family member holds Novo Nordisk stock. All other authors declare no competing interests.

## Additional information

Simon Birk Kjær Jensen[1], Charlotte Janus[1], Julie Rehné Lundgren[1], Christian Rimer Juhl[1], Rasmus Michael Sandsdal [1], Lisa Møller Olsen[1], Anne Andresen[1], Signe Amalie Borg [1], Ida Christine Jacobsen [1], Graham Finlayson[2], Bente Merete Stallknecht[1], Jens Juul Holst [1,3], Sten Madsbad [4] & Signe Sørensen Torekov [1]✉

[1]Department of Biomedical Sciences, Faculty of Health and Medical Sciences, University of Copenhagen, Copenhagen, Denmark. [2]School of Psychology, Faculty of Medicine and Health, University of Leeds, Leeds, UK. [3]NNF Center for Basic Metabolic Research, Faculty of Health and Medical Sciences, University of Copenhagen, Copenhagen, Denmark. [4]Department of Endocrinology, Hvidovre University Hospital, Hvidovre, Denmark. ✉e-mail: torekov@sund.ku.dk

