## [Peer Review File · Nature Communications]

Title: Eating and physical activity-related outcomes in an exploratory analysis of a randomized controlled trial for weight loss maintenance with exercise, liraglutide, or both combinedREVIEWER COMMENTS

Reviewer #1 (Remarks to the Author):

In this manuscript, Jensen and colleagues report the results of a placebo-controlled randomized clinical trial aimed at preventing relapse of eating and sedentary behaviors after an initial weight loss, using an exercise intervention, liraglutide, and the combination of exercise and liraglutide. The findings are potentially interesting; however, the focus on the per-protocol population and discrepancies between the study protocol and the analyses in the manuscript are of concern.

Major

1. The primary analyses should be in the ITT population (see PMID: 10822117). Excluding the 65 participants who did not adhere to study interventions or who did not complete assessments may bias the findings in favor of the intervention.
2. There are a number of discrepancies between the protocol and the analyses presented. Body weight is listed as the primary outcome but is described in one sentence on page 5. There is a power calculation for the comparison of liraglutide vs. placebo with a sample size of 30 per group, but there are 4 groups in the analysis.
3. The journal's instructions to authors state that clinical research should follow CONSORT guidelines for reporting. There is no participant flow diagram, and there are other elements of the CONSORT guidelines that are not followed (e.g., description of how the sample size was determined, reporting of harms).

Minor

1. There should be some discussion of how missing data were handled during the analysis.
2. Please provide justification/reference regarding the choice to include accelerometer files if data were available >16 hours per day, 3 days pwer week.
3. The discussion (page 12, line 259) comments on the multiple repeated measures as a strength of this study but these results are not presented except descriptively in Figure 3. The analyses focus on change from the end of weight loss to 1 year later.
4. Page 3, line 43: from a Pubmed search, it appears that eating and sedentary behavior during weight loss maintenance have previously been characterized in certain settings, so this should probably be softened a bit.
5. Figure 4 implies that sedentary time decreased during the LCD phase, but the discussion states there were no acute changes in sedentary time. Is there a reason why sedentary time would decrease during LCD?
6. In Figure 3, there are peaks in energy expenditure and duration between 30-35 weeks after the start of the intervention. Is there a potential explanation for this?
7. Minor grammatical issues and typos: e.g., "extend" instead of "extent," missing "%" on page 8, line 156.

Reviewer #2 (Remarks to the Author):

This manuscript presents an exploratory analysis of appetite, eating behaviour and physical activity in completers of a 4-arm RCT of liraglutide and/or exercise for maintenance of weight loss over 12 months. The study design and methodology are strong and described more comprehensively elsewhere, along with the primary outcomes.

The noteworthy results of this analysis are that appetite and sedentary behaviour increased with placebo and did not change significantly during follow-up in liraglutide-treated groups. These findings are interesting and although the effects of weight loss, GLP1RA and exercise on appetite have been reported previously, the present study is novel in its description of the effects of GLP1RA and exercise separately and in combination.

My concerns mainly relate to the overstatement of results in some areas, and the wording throughout the discussion that suggests that the successful weight maintenance is caused by the appetite, eating behaviour and physical activity results (which is not directly examined here).

Introduction:

The aims of this analysis should be stated

Line 46 suggest delete "inexpedient" as it sounds somewhat stigmatising

Results:

Table 2: It is not clear why the LCD column presents data for the whole group that completed the weight loss phase (n=195), whereas the remaining columns show only the completers of the 12 month follow-up (though the numbers in treatment arms add up to n=131 rather than n=130). It would make more sense to show the post-LCD data for the completers, or to present (not necessarily in the same table) the differences in eating behaviours and appetite ratings between the completers and the remaining 33% of post-LCD participants who were excluded. I have the same comment about Table 3.

Please change the section heading in line 150 ("exercise does not increase appetite"), which is not consistent with the results presented, or with the (accurate) sentence (line 160) "thus, participants who exercised experienced increased appetite"

The section heading line 165 ("combination L+E improve eating behaviour" and corresponding sentence in the discussion line 194) also seems overstated compared with the results presented below it.

Although the change in cognitive restraint was higher in the L+E than the placebo group, there was no significant change in TFEQ scores in either group.

Discussion:

In several areas (e.g. line 190, 192, 200, 207, 237, 265, 268) it is stated that the findings presented are causative or contributory to the weight changes (primary outcome). While it is likely that this is the case, causality cannot be determined here, and even associations between the behaviours and the weight outcome were not presented in this analysis. Therefore, this wording needs to be amended.

Line 223 states that liraglutide's effect on weight maintenance involves inhibition of the preference for high-fat foods observed in the placebo group, but this is contradictory to the results in Table 2 showing that wanting and liking of high fat savoury foods decreased also in the placebo group to a magnitude not significantly different from the liraglutide group. Could the authors comment on what they think is the explanation for this?

Line 272 (and line 32 abstract) the conclusion that targeting both eating and sedentary behaviour seem

crucial for preventing weight regain is not supported by the findings from this analysis or the primary outcome paper showing that weight regain was prevented also in the liraglutide and exercise alone groups.

Other:

Line 63 – this study showed increased (not reduced) fullness with liraglutide

There are a few typographical errors (line 112, 159, 161 “extent”; line 127 “table 3”)

Reviewer #3 (Remarks to the Author):

Overall very good manuscript, building on previous work done by this group. The format of the manuscript should be more in line with research articles. Intro, Methods, Results, Discussion. I appreciate that the group wanted to break it up into section based on each question and including methods and results under each heading. This is not the standard, and I would revert to the standard. Under Results, it can be broken up into each question. I see that methods are below the discussion and conclusion, they should be moved to the correct position. I have included my comments in the document. I disagree with aspects of the conclusion that exercise does not increase appetite.

REVIEWER COMMENTS

Reviewer #1 (Remarks to the Author):

In this manuscript, Jensen and colleagues report the results of a placebo-controlled randomized clinical trial aimed at preventing relapse of eating and sedentary behaviors after an initial weight loss, using an exercise intervention, liraglutide, and the combination of exercise and liraglutide. The findings are potentially interesting; however, the focus on the per-protocol population and discrepancies between the study protocol and the analyses in the manuscript are of concern.

Author response: We thank Reviewer #1 for reviewing our manuscript. We have carefully responded to each of the comments made in the point-by-point response below and amended/clarified the manuscript accordingly.

Major

1. The primary analyses should be in the ITT population (see PMID: 10822117). Excluding the 65 participants who did not adhere to study interventions or who did not complete assessments may bias the findings in favor of the intervention.

Author response:

We agree with the reviewer that analyzing the intention-to-treat population is important to provide an unbiased treatment estimate due to the effect of randomization. In this manuscript, we provide both intention-to-treat (Table 4) and per-protocol analysis (Table 2 and 3) as defined in the published statistical analysis plan (Lundgren et al. 2021). The per-protocol analysis reflects those who completed the study and adhered to the allocated treatments and shows the effects when the treatments are taken optimally. We performed the per-protocol analysis to get a better mechanistic understanding of the behavioral changes that occur after weight loss and how the different actual performed treatments may affect these. We also performed the analyses on the full data set including all randomized participants (intention-to-treat population) (previous Tables S1 and S2). In total, 36 participants completed the study without adequate adherence to allocated treatment. Including these participants in the analyses generally showed the same picture as that presented for the per-protocol population. Both intention-to-treat and per-protocol analysis are important and should be shown in the main manuscript. We have therefore moved previous Tables S1 and S2 to the main manuscript as Table 4. Furthermore, we have presented intention-to-treat analyses in Figures S3 and S4. With analyses made on both the per-protocol and intention-to-treat populations, the readers are provided with a good background to interpret the intervention effects. As stated in the protocol, we specified to do the analyses on both populations. The definition of the per-protocol population was predefined in the statistical analysis plan, which is also now clearly stated in the manuscript. We have now made the intention-to-treat analyses more visible in the main text, tables, and figures and we included a discussion of the use of per-protocol analyses in discussion sections.

Changes in manuscript:

Paragraph added in the *results*: “In the intention-to-treat population (including all randomized participants regardless of adherence), the self-reported daily sitting time was lower in the combination group compared with placebo, whereas the accelerometer measured sedentary time was not significantly decreased compared with placebo (Table 4). This discrepancy was driven by

increased sedentary time in some participants randomized to the combination group, who did not adhere to the treatment regimen (Figure S4).” (Page 9 line 193).

And paragraph about limitations added in the *discussion*: “We have presented both the intention-to-treat population and the per-protocol population, the latter meaning those who completed the study and adhered to the allocated treatment. The per-protocol analysis reflects the effects when the treatments are taken optimally, and we did this to get a better mechanistic understanding of the behavioral changes that occur after weight loss and how the different actual performed treatments may affect these. A limitation of this approach is the introduction of potential subset selection bias that cannot be distinguished from the treatment effect (Lachin 2000). We also performed the analyses on the full data set including all randomized participants (intention-to-treat population). In total, 36 participants completed the study without adequate adherence to allocated treatment. Including these participants in the analyses (Table 4) generally showed the same picture as those presented for the per-protocol population.” (Page 13 line 295).

Paragraph added in the *statistical analyses*: “We performed the analyses on both the per-protocol and intention-to-treat population. The per-protocol population includes participants with adequate adherence to study interventions, and was pre-defined (see Lundgren et al. 2021) for the exercise intervention as meeting at least 75% of WHO’s recommendations on physical activity for health in adults: at least 150 minutes of moderate-intensity aerobic physical activity, or 75 minutes of vigorous-intensity aerobic physical activity, or an equivalent combination of both (Bull et al. 2020). For study medication, per-protocol was defined as having administered 2.4 or 3.0 mg/day subcutaneous liraglutide/placebo for at least 75% of the intervention period. The intention-to-treat population was defined as all randomized participants regardless of adherence” (Page 19 line 427).

2. There are a number of discrepancies between the protocol and the analyses presented. Body weight is listed as the primary outcome but is described in one sentence on page 5. There is a power calculation for the comparison of liraglutide vs. placebo with a sample size of 30 per group, but there are 4 groups in the analysis.

Author response: Body weight was the primary outcome of the study and has been published previously (Lundgren et al. 2021), as stated in the introduction (page 4, line 76) and in the methods section (page 14, line 320). Sample size was calculated based on a between-group difference in body weight of 4 kg as described in the published statistical analysis plan (available at [clinicaltrials.gov, NCT04122716](https://clinicaltrials.gov/ct2/show/study/NCT04122716)). In the present paper, we present results on secondary outcomes. We have now stated more clearly that the primary endpoint has been published while in the present paper, we present results on pre-specified secondary outcomes.

Changes in manuscript:

Paragraph in the *results* revised from:

“The low-calorie diet induced a weight loss of 13 kg.”

To:

“As described previously (Lundgren et al. 2021), the low-calorie diet induced a weight loss of 13.1 kg.” (Page 5 line 106).

And paragraph added in the *statistical analyses*:

“The sample size was calculated based on change in body weight, which was the primary outcome and has been published (Lundgren et al. 2021). It was estimated that at least 30 participants in each group would be necessary to detect a clinical relevant 4 kg difference between any of the four treatment groups. In this paper, we report results on pre-specified secondary outcomes.” (Page 19 line 421).

3. The journal’s instructions to authors state that clinical research should follow CONSORT guidelines for reporting. There is no participant flow diagram, and there are other elements of the CONSORT guidelines that are not followed (e.g., description of how the sample size was determined, reporting of harms).

Author response: We agree and apologize for that. We have now included a CONSORT flow diagram in the supplementary appendix (Figure S1). Description of sample size calculations and reporting of harms, as well as other elements of the CONSORT guidelines (e.g. details on randomization, blinding, dates of recruitment) have been included.

Changes in manuscript: Reporting of harms added in the *Results* section:

“*Harms*

Harms in the study have been reported in detail elsewhere (Lundgren et al. 2021). In brief, gastrointestinal adverse events were more frequent in the liraglutide and combination group than in the exercise and placebo groups. This included nausea, diarrhea, and vomiting. The number of participants reporting decreased appetite during the trial was 2 (4%) in the placebo group, 4 (8%) in the exercise group, 18 (37%) in the liraglutide group, and 16 (33%) in the combination group. A total of 16 (8%) serious adverse events were reported, of which 5 (3%) led to discontinuation of study medication.” (Page 10 line 209).

Paragraph in *Methods* revised from:

“The reported results were part of a randomized placebo-controlled trial (EudraCT number, 2015-005585-32; clinicaltrials.gov number, NCT04122716). The study protocol and primary outcome (body weight) have been published (Jensen et al. 2019; Lundgren et al. 2021). The study was approved by the local ethics committee and the Danish Medicines Agency and was carried out in accordance with the principles of the Declaration of Helsinki and ICH Good Clinical Practice guidelines.”

To:

“The reported results were part of a randomized placebo-controlled, 2-by-2 factorial trial (EudraCT number, 2015-005585-32; clinicaltrials.gov number, NCT04122716). The study protocol, statistical analysis plan, and primary outcome (body weight) have been published (Jensen et al. 2019; Lundgren et al. 2021). The study was approved by the local ethics committee (H-16027082) and the Danish Medicines Agency and was carried out in accordance with the principles of the Declaration of Helsinki and ICH Good Clinical Practice guidelines.” (Page 14 line 320).

Paragraph in *methods* revised from:

“Men and women (n=215) were included from August 2016 at Hvidovre University Hospital and Department of Biomedical Sciences, University of Copenhagen, Denmark.”

To:

“Men and women (n=215) were included from August 2016 to September 2018 at Hvidovre University Hospital and Department of Biomedical Sciences, University of Copenhagen, Denmark. Last participant’s last visit was November 2019.” (Page 15 line 328).

Paragraph in *Methods* revised from:

“Participants with weight loss $\geq 5\%$ (n=195) were then randomly assigned, in a 1:1:1:1 ratio, to a one-year weight loss maintenance phase with either exercise+placebo, liraglutide, a combination of exercise and liraglutide, or placebo. The starting dose of liraglutide (or volume-matched placebo) was 0.6 mg/day with 0.6 mg weekly increments until 3.0 mg/day or the highest dose at which participants did not have unacceptable adverse events. The participants and the investigators were blinded according to liraglutide or placebo treatment.”

To:

“Participants with weight loss $\geq 5\%$ (n=195) were then randomly assigned, in a 1:1:1:1 ratio (with stratification according to gender and age group (<40 years and ≥ 40 years), to a one-year weight loss maintenance phase with either exercise+placebo, liraglutide, a combination of exercise and liraglutide, or placebo. Assignment of participants to treatment was performed by a study nurse based on a randomization list provided by Novo Nordisk. The starting dose of liraglutide (or volume-matched placebo) was 0.6 mg/day with 0.6 mg weekly increments until 3.0 mg/day or the highest dose at which participants did not have unacceptable adverse events. The participants and the investigators were blinded according to liraglutide or placebo treatment until the analyses of the primary outcome were complete.” (Page 15 line 334).

Minor

1. There should be some discussion of how missing data were handled during the analysis.

Author response: We agree. The missing data were handled according to the published statistical analysis plan (https://clinicaltrials.gov/ProvidedDocs/16/NCT04122716/SAP_001.pdf). The analyses were performed under the assumption that data was missing at random. Under this assumption the applied mixed linear model imputes missing values using the measured values using maximum likelihood estimation.

Changes in manuscript: Paragraph included in the *Statistical analyses*:

“For all analyses, missing data was assumed to be missing at random. Reasons for missingness can be seen in the flow diagram (Fig. S1).” (Page 19 line 435).

2. Please provide justification/reference regarding the choice to include accelerometer files if data were available >16 hours per day, 3 days per week.

Author response: We wanted to include as much data as possible in the analyses that would still be representative of a behavior. The 3-day cut-off has previously been used in accelerometer analyses of physical activity and is based on simulation analyses of complete datasets (>29,000 participant data; 7 days of data), simulating the effects of different amounts of available data. These analyses

have shown that at least 3 days are needed to be within 10% of the true stable seven-day measure (Doherty et al. 2017). This has been included in the paper as reference.

Changes in manuscript: Paragraph in the *methods* revised from:

“Files were included in the analysis if data were available for >16 hours/day on at least three days of the week.”

To:

“Files were included in the analysis if data were available for >16 hours/day on at least three days of the week, as this has been shown to be representative of a full week (Doherty et al. 2017).” Page 18 line 409:

3. The discussion (page 12, line 259) comments on the multiple repeated measures as a strength of this study but these results are not presented except descriptively in Figure 3. The analyses focus on change from the end of weight loss to 1 year later.

Author response: We agree that the repeated measures should be presented more clearly.

Accelerometer data was collected five times in total during the study. These measures were

included in the statistical model to measure changes from after weight loss to after one year.

We have included a graph with changes in sedentary time over time, to visualize the changes in sedentary time during the study (Figure 2A).

4. Page 3, line 43: from a Pubmed search, it appears that eating and sedentary behavior during weight loss maintenance have previously been characterized in certain settings, so this should probably be softened a bit.

Author response: We acknowledge that this could be softened and we have revised it accordingly.

Changes in manuscript: Paragraph in the *introduction* revised from:

“However, the actual eating and sedentary behavior during weight loss maintenance treatment after weight loss have not been characterized.”

To:

“However, the actual eating and sedentary behavior also seem to be important regulators of body weight (Christensen et al. 2018; Jebb and Moore 1999). How different weight loss maintenance strategies affect eating and sedentary behavior has not been characterized.” (Page 3 line 42).

5. Figure 4 implies that sedentary time decreased during the LCD phase, but the discussion states there were no acute changes in sedentary time. Is there a reason why sedentary time would decrease during LCD?

Author response: We stated that there was no change in sedentary time during LCD, as the 12 min/day change was not statistically significant (95%CI, -27 to 2) and it may therefore be a coincidence with the non-significant decrease sedentary time during LCD. We have clarified this.

Changes in manuscript: Paragraph in the *discussion* revised from:

“In our study, there were no acute changes in physical activity or sedentary time during the low-calorie diet.”

To:

“In our study, there were no acute changes in physical activity during the low-calorie diet, and a non-significant change in sedentary time of -12 min/day. (Page 11 line 236).

6. In Figure 3, there are peaks in energy expenditure and duration between 30-35 weeks after the start of the intervention. Is there a potential explanation for this?

Author response: We have looked through the individual participants’ data to find an explanation. It appears that a few participants had a large exercise volume at this time point, which was enough to manifest in the group mean. It seems that these instances of high exercise volume were during active vacation (e.g. cycling vacation). Participants were recruited continuously, meaning that week 30-35 was not the same time of the year for participants. Thus, the peak in energy expenditure appears to be a coincidence.

7. Minor grammatical issues and typos: e.g., “extend” instead of “extent,” missing “%” on page 8, line 156.

Author response: We apologize and have revised it accordingly.

Reviewer #2 (Remarks to the Author):

This manuscript presents an exploratory analysis of appetite, eating behaviour and physical activity in completers of a 4-arm RCT of liraglutide and/or exercise for maintenance of weight loss over 12 months.

The study design and methodology are strong and described more comprehensively elsewhere, along with the primary outcomes.

The noteworthy results of this analysis are that appetite and sedentary behaviour increased with placebo and did not change significantly during follow-up in liraglutide-treated groups. These findings are interesting and although the effects of weight loss, GLP1RA and exercise on appetite have been reported previously, the present study is novel in its description of the effects of GLP1RA and exercise separately and in combination.

My concerns mainly relate to the overstatement of results in some areas, and the wording throughout the discussion that suggests that the successful weight maintenance is caused by the appetite, eating behaviour and physical activity results (which is not directly examined here).

Author response: We thank Reviewer #2 for reviewing our manuscript. We acknowledge the concern on claiming causal inferences, and have addressed and amended this throughout the manuscript; see below for specific changes.

Introduction:

The aims of this analysis should be stated

Author response: We have clarified the aim in the introduction.

Changes in manuscript: Paragraph added in the *introduction*:

“Thus, the aim of the present study was to investigate changes in appetite, eating and sedentary behavior, and non-exercise physical activity during one-year weight loss maintenance with

moderate-to-vigorous intensity exercise, liraglutide 3.0 mg, or a combination of both, compared with placebo after an initial diet-induced weight loss.” (Page 4 line 80).

Line 46 suggest delete “inexpedient” as it sounds somewhat stigmatizing

Author response: We agree and have changed the wording.

Results:

Table 2: It is not clear why the LCD column presents data for the whole group that completed the weight loss phase (n=195), whereas the remaining columns show only the completers of the 12 month follow-up (though the numbers in treatment arms add up to n=131 rather than n=130). It would make more sense to show the post-LCD data for the completers, or to present (not necessarily in the same table) the differences in eating behaviours and appetite ratings between the completers and the remaining 33% of post-LCD participants who were excluded. I have the same comment about Table 3.

Author response: We agree and have changed Tables 2 and 3 to include the per-protocol population, also during the LCD. In a separate table (Table 4), we have included data for the whole population (intention-to-treat) during LCD and one year of subsequent treatment.

Please change the section heading in line 150 (“exercise does not increase appetite”), which is not consistent with the results presented, or with the (accurate) sentence (line 160) “thus, participants who exercised experienced increased appetite”

Author response: We agree and have changed accordingly.

Changes in manuscript: Subheading in the *results* revised from:

“Exercise does not increase appetite despite increased exercise energy expenditure and maintained weight loss”

To:

“Exercise increases appetite similar to placebo, despite increased exercise energy expenditure and maintained weight loss”. (Page 8 line 159).

The section heading line 165 (“combination L+E improve eating behaviour” and corresponding sentence in the discussion line 194) also seems overstated compared with the results presented below it. Although the change in cognitive restraint was higher in the L+E than the placebo group, there was no significant change in TFEQ scores in either group.

Author response: We agree and have changed the heading and revised the discussion to avoid overstatements of the results.

Changes in manuscript: Subheading in the *results* revised from:

“The combination of liraglutide and exercise improve eating behavior and reduce sedentary time during weight maintenance”.

To:

“The combination of liraglutide and exercise improves cognitive restraint and reduces sedentary time during weight maintenance”. (Page 8 line 174).

And paragraph in the *discussion* revised from:

“The combination of exercise and GLP-1 RA improved both cognitive restraint, reflecting a conscious restriction of food intake as well as prevented the increase in sedentary behavior, thereby facilitating additional weight loss.”

To:

“Compared with placebo treatment, the combination of exercise and GLP-1 RA improved both cognitive restraint, reflecting a conscious restriction of food intake, as well as prevented the increase in sedentary behavior, which may have facilitated the additional weight loss.” (Page 10 line 223).

Discussion:

In several areas (e.g. line 190, 192, 200, 207, 237, 265, 268) it is stated that the findings presented are causative or contributory to the weight changes (primary outcome). While it is likely that this is the case, causality cannot be determined here, and even associations between the behaviours and the weight outcome were not presented in this analysis. Therefore, this wording needs to be amended.

Author response: Thank you for pointing this out and we agree. We have therefore revised throughout the manuscript and removed wording on causal inferences. Further, we have included a multiple linear regression analysis to test for associations between the behaviors and change in body weight (see Table 5).

Changes in manuscript: Examples of changes are:

Paragraph added in the *results*:

“Increased cognitive restraint and moderate-to-vigorous intensity physical activity are associated with maintained weight loss

We performed a multiple regression analysis in order to test if changes in the investigated behaviors were associated with changes in body weight during the weight maintenance phase (Table 5).

Increased cognitive restraint and amount of moderate-to-vigorous intensity physical activity were associated with less regain during weight maintenance ($P=0.003$ and $P=0.02$, respectively).

Increased uncontrolled eating tended to be associated with more weight regain ($P=0.07$).” (Page 9 line 200).

And paragraph in the *discussion* revised from:

Change from: “Placebo treatment after weight loss was associated with increased appetite and sedentary time contributing to the weight regain after weight loss. Liraglutide treatment after weight loss prevented the increase in appetite, thereby leading to maintained weight loss.”

To: “Placebo treatment after weight loss was associated with increased appetite and sedentary time which may have contributed to the weight regain. Liraglutide treatment after weight loss prevented the increase in appetite, which may have contributed to the maintained weight loss.” (Page 10 line 219).

And paragraph in the *discussion* revised from:

“After weight loss, in the placebo group, the perception of postprandial hunger, prospective food consumption and wanting of food items that were sweet and high in fat increased, while the

perception of postprandial satiety and fullness decreased, which contributed to a positive energy balance and thereby a weight regain of almost 50% of the initial weight loss within one year”.

To:

“After weight loss, the perception of postprandial hunger, prospective food consumption, and wanting of food items that were sweet and high in fat increased in the placebo group, while the perception of postprandial satiety and fullness decreased. Increased appetite promotes increased energy intake and weight regain (Polidori et al. 2016; Hall and Kahan 2018), and is therefore likely to have contributed to a positive energy balance and thereby the weight regain of almost 50% of the initial weight loss within one year observed in the placebo group”. (Page 11 line 228).

And paragraph added in the *discussion*: “Increased moderate-to-vigorous intensity physical activity was associated with less weight regain in the whole study population, in support of physical activity being important for weight maintenance.” (Page 12 line 262).

And paragraph in the *discussion* revised from:

“Thus, increased overall physical activity in the exercise group was not associated with a further increase in appetite, which, together with the increased energy expenditure from the exercise, seem to have contributed to successful weight maintenance.”

To:

“Thus, increased overall physical activity in the exercise group was not associated with a further increase in appetite compared with placebo, which, together with the increased energy expenditure from the exercise, may have contributed to successful weight maintenance.” (Page 12 line 267).

And paragraph in the *discussion* revised from:

“The combination group also improved the cognitive restraint score, reflecting an intent to restrict energy intake to control body weight which also seems to be an important factor in weight loss maintenance (Graham et al. 2014; Keränen et al. 2009).”

To:

The combination group also improved the cognitive restraint score, reflecting an intent to restrict energy intake to control body weight. Performing linear regression analysis on the study group as a whole, we found that increased cognitive restraint was associated with less weight regain during weight maintenance (Table 5). Cognitive restraint has previously been proposed to be an important factor in weight loss maintenance (Graham et al. 2014; Keränen et al. 2009).” (Page 13 line 276).

And paragraph added in the *statistical analyses*: “Multiple linear regression analysis was performed to assess potential associations between changes in the investigated outcomes and change in body weight during the weight loss maintenance intervention. Age, gender, intervention group, and baseline body weight were included as covariates in the model.” (Page 19 line 438).

Line 223 states that liraglutide’s effect on weight maintenance involves inhibition of the preference for high-fat foods observed in the placebo group, but this is contradictory to the results in Table 2 showing that wanting and liking of high fat savoury foods decreased also in the placebo group to a

magnitude not significantly different from the liraglutide group. Could the authors comment on what they think is the explanation for this?

Author response: We agree that the changes in food preferences do not significantly differ between liraglutide and placebo. Although it seems that the changes in food preferences generally favor liraglutide, we cannot claim that liraglutide decreases high-fat foods compared with placebo. We have therefore reworded throughout the section. The reason for the observed decrease in preferences for high fat savory foods in the placebo group is not clear, but could (for wanting at least) be related to a normalization of the change that occur during the low-calorie diet. The meal replacement products were generally sweet in taste, and we speculate that this has driven some of the acute decrease in preferences for sweet and increase for savory observed after the low-calorie diet.

Changes in manuscript: Paragraph in the *discussion* revised from:

“Together, the durable effect of liraglutide in maintaining the diet-induced weight loss seem to involve inhibition of the increased appetite and preference for high-fat foods that were observed in the placebo group, which persisted for at least a year.”

To:

“Together, the durable effect of liraglutide in maintaining the diet-induced weight loss seems to involve inhibition of the increased appetite that was observed in the placebo group, which persisted for at least a year.” (Page 12 line 253).

Line 272 (and line 32 abstract) the conclusion that targeting both eating and sedentary behaviour seem crucial for preventing weight regain is not supported by the findings from this analysis or the primary outcome paper showing that weight regain was prevented also in the liraglutide and exercise alone groups.

Author response: We agree that this wording needs to be amended to more accurately reflect the actual results. The combination treatment led to further weight loss beyond the initial weight loss, thus it seems more effective in preventing weight regain compared with liraglutide and exercise alone which only maintained weight loss.

Changes in manuscript: Paragraph in the *abstract* revised from:

“The combination of exercise and liraglutide improved cognitive restraint by 21%, reflecting a conscious restriction of food intake, and decreased sedentary time by 41 min/day compared with placebo, thereby facilitating additional weight loss. Targeting both appetite and sedentary behavior seems crucial for preventing weight regain.”

To:

“The combination of exercise and liraglutide improved cognitive restraint by 21%, reflecting a conscious restriction of food intake, and decreased sedentary time by 41 min/day compared with placebo, which may have facilitated additional weight loss. Targeting both eating and sedentary behavior seems the most effective for preventing weight regain.” (Page 2 line 29).

And paragraph in the *discussion* revised from:

“The combination of exercise and liraglutide improved cognitive restraint and decreased sedentary behavior compared with placebo, thereby facilitating additional weight loss. Thus, targeting both eating and sedentary behavior seems crucial for preventing weight regain.”

To:

“The combination of exercise and liraglutide improved cognitive restraint and decreased sedentary behavior compared with placebo, which may have facilitated the additional weight loss. Thus, targeting both eating and sedentary behavior seems the most effective approach for preventing weight regain after weight loss. (Page 14 line 313).

Other:

Line 63 – this study showed increased (not reduced) fullness with liraglutide

Author response: We apologize and have corrected it.

There are a few typographical errors (line 112, 159, 161 “extent”; line 127 “table 3”)

Author response: We have revised.

Reviewer #3 (Remarks to the Author):

Overall very good manuscript, building on previous work done by this group.

Author response: We thank Reviewer #3 for reviewing our manuscript.

The format of the manuscript should be more in line with research articles. Intro, Methods, Results, Discussion. I appreciate that the group wanted to break it up into section based on each question and including methods and results under each heading. This is not the standard, and I would revert to the standard. Under Results, it can be broken up into each question. I see that methods are below the discussion and conclusion, they should be moved to the correct position.

Author response: We acknowledge that the format does not follow the standard format. However, the placement of methods after discussion was done because it is stated in the Nature Communications format guidelines for articles (<https://www.nature.com/ncomms/submit/article>) that “*the main text of an Article should begin with a section headed Introduction ... followed by sections headed Results, Discussion (if appropriate) and Methods (if appropriate).*” Thus, we followed the journal’s style recommendations. However, we are happy to change format if required.

I have included my comments in the document.

Reviewer #3 comments in the document:

Where is the limitations section. There are a number of limitations, including the fact that it was open labelled, non blinded.

Author response: We agree and have now included a limitations section in the discussion. Further, we have clarified that the study medication was blinded during treatment and was unblinded after primary outcomes analyses.

Changes in manuscript: Paragraph added in the *discussion*:

“It is a potential limitation is that we cannot prove causal inferences that the measured outcomes are the reason for the weight changes, although this seems probable. We have presented both the intention-to-treat population and the per-protocol population, the latter meaning those who completed the study and adhered to the allocated treatment. The per-protocol analysis reflects the effects when the treatments are taken optimally, and we did this to get a better mechanistic understanding of the behavioral changes that occur after weight loss and how the actual performed treatments may affect these. A limitation of this approach is the introduction of potential subset selection bias that cannot be distinguished from the treatment effect (Lachin 2000). However, we also performed the analyses on the full data set including all randomized participants (intention-to-treat population). In total, 36 participants completed the study without adequate adherence to allocated treatment. Including these participants in the analyses (Table 4) generally showed the same picture as that presented for the per-protocol population.” (Page 13 line 294).

And paragraph added in the *methods*:

“The participants and the investigators were blinded with respect to liraglutide or placebo treatment until the analyses of the primary outcome were complete.” (Page 15 line 341).

I disagree with aspects of the conclusion that exercise does not increase appetite.

Author response: We agree and have made it clear that appetite is increased in the exercise group similar to the placebo group despite the increased energy expenditure in the exercise group.

Changes in manuscript: Paragraph in the *discussion* revised from:

“Exercise after weight loss did not increase appetite or sedentary behavior despite increased exercise energy expenditure and maintained weight loss.”

To:

“Exercise after weight loss led to a similar increase in appetite as the placebo group, despite increased exercise energy expenditure and maintained weight loss.” (Page 14 line 311).

References

- Bull, Fiona C., Salih S. Al-Ansari, Stuart Biddle, Katja Borodulin, Matthew P. Buman, Greet Cardon, Catherine Carty, et al. 2020. "World Health Organization 2020 Guidelines on Physical Activity and Sedentary Behaviour." *British Journal of Sports Medicine* 54 (24): 1451–62. <https://doi.org/10.1136/bjsports-2020-102955>.
- Christensen, Bodil Just, Eva Winning Iepsen, Julie Lundgren, Lotte Holm, Sten Madsbad, Jens Juul Holst, and Signe Sørensen Torekov. 2018. "Instrumentalization of Eating Improves Weight Loss Maintenance in Obesity." *Obesity Facts* 10 (6): 633–47. <https://doi.org/10.1159/000481138>.
- Doherty, Aiden, Dan Jackson, Nils Hammerla, Thomas Plötz, Patrick Olivier, Malcolm H. Granat, Tom White, et al. 2017. "Large Scale Population Assessment of Physical Activity Using Wrist Worn Accelerometers: The UK Biobank Study." *PLoS ONE* 12 (2): e0169649. <https://doi.org/10.1371/journal.pone.0169649>.
- Graham, Alexis L., Marci E. Gluck, Susanne B. Votruba, Jonathan Krakoff, and Marie S. Thearle. 2014. "Perseveration Augments the Effects of Cognitive Restraint on Ad Libitum Food Intake in Adults Seeking Weight Loss." *Appetite* 82 (November): 78–84. <https://doi.org/10.1016/j.appet.2014.07.008>.
- Hall, Kevin D., and Scott Kahan. 2018. "Maintenance of Lost Weight and Long-Term Management of Obesity." *Medical Clinics of North America*. W.B. Saunders. <https://doi.org/10.1016/j.mcna.2017.08.012>.
- Jebb, Susan A., and Melanie S. Moore. 1999. "Contribution of a Sedentary Lifestyle and Inactivity to the Etiology of Overweight and Obesity: Current Evidence and Research Issues." *Medicine and Science in Sports and Exercise* 31 (11 SUPPL.). <https://doi.org/10.1097/00005768-199911001-00008>.
- Jensen, Simon Birk Kjær, Julie Rehné Lundgren, Charlotte Janus, Christian Rimer Juhl, Lisa Møller Olsen, Mads Rosenkilde, Jens Juul Holst, Bente Merete Stallknecht, Sten Madsbad, and Signe Sørensen Torekov. 2019. "Protocol for a Randomised Controlled Trial of the Combined Effects of the GLP-1 Receptor Agonist Liraglutide and Exercise on Maintenance of Weight Loss and Health after a Very Low-Calorie Diet." *BMJ Open* 9 (11): e031431. <https://doi.org/10.1136/bmjopen-2019-031431>.
- Keränen, Anna Maria, Markku J. Savolainen, Annakaisa H. Reponen, Mona Lisa Kujari, Sari M. Lindeman, Risto S. Bloigu, and Jaana H. Laitinen. 2009. "The Effect of Eating Behavior on Weight Loss and Maintenance during a Lifestyle Intervention." *Preventive Medicine* 49 (1): 32–38. <https://doi.org/10.1016/j.ypmed.2009.04.011>.
- Lachin, John M. 2000. "Statistical Considerations in the Intent-to-Treat Principle." *Controlled Clinical Trials* 21 (3): 167–89. [https://doi.org/10.1016/S0197-2456\(00\)00046-5](https://doi.org/10.1016/S0197-2456(00)00046-5).
- Lundgren, Julie R., Charlotte Janus, Simon B.K. Jensen, Christian R. Juhl, Lisa M. Olsen, Rasmus M. Christensen, Maria S. Svane, et al. 2021. "Healthy Weight Loss Maintenance with Exercise, Liraglutide, or Both Combined." *New England Journal of Medicine* 384 (18): 1719–30. <https://doi.org/10.1056/NEJMoa2028198>.
- Polidori, David, Arjun Sanghvi, Randy J. Seeley, and Kevin D. Hall. 2016. "How Strongly Does Appetite Counter Weight Loss? Quantification of the Feedback Control of Human Energy Intake." *Obesity* 24 (11): 2289–95. <https://doi.org/10.1002/oby.21653>.

REVIEWERS' COMMENTS

Reviewer #1 (Remarks to the Author):

My comments have been addressed - thank you.

Reviewer #2 (Remarks to the Author):

The authors have addressed my previous concerns well - thank you.

I have a few minor suggestions:

1. error line 136 - table 3
2. line 142 "relapse" - I suggest "reduction" might be clearer
3. line 238-9 suggest delete (or modify to soften) the sentence "Thus, increased sedentary time also seems to contribute to weight regain after weight loss" because sedentary time was not associated with weight regain in the multiple linear regression analysis.
4. line 86 and 311 "Exercise after weight loss led to a similar increase in appetite as the placebo group, despite increased exercise energy expenditure..." - the authors' point might be clearer if wording indicated that the exercise only group did not experience a greater increase in appetite than the placebo group

REVIEWERS' COMMENTS

Reviewer #1 (Remarks to the Author):

My comments have been addressed - thank you.

Reviewer #2 (Remarks to the Author):

The authors have addressed my previous concerns well - thank you.

Author Response: We thank reviewer 1 and 2 for reviewing the manuscript and giving valuable comments that have improved the manuscript.

I have a few minor suggestions:

1. error line 136 - table 3

Author Response: This has been corrected.

2. line 142 "relapse" - I suggest "reduction" might be clearer

Author Response: We agree and have changed accordingly.

3. line 238-9 suggest delete (or modify to soften) the sentence "Thus, increased sedentary time also seems to contribute to weight regain after weight loss" because sedentary time was not associated with weight regain in the multiple linear regression analysis.

Author Response: We agree and have changed to: "Thus, increased sedentary time may contribute to weight regain after weight loss."

4. line 86 and 311 "Exercise after weight loss led to a similar increase in appetite as the placebo group, despite increased exercise energy expenditure..." - the authors' point might be clearer if wording indicated that the exercise only group did not experience a greater increase in appetite than the placebo group

Author Response: We agree and have revised to: "Exercise after weight loss did not lead to a greater increase in appetite than the placebo group, despite increased exercise energy expenditure and maintained weight loss."